# SARS-CoV-2 Spike Protein Intensifies Cerebrovascular Complications in Diabetic hACE2 Mice through RAAS and TLR Signaling Activation

**DOI:** 10.3390/ijms242216394

**Published:** 2023-11-16

**Authors:** Faith N. Burnett, Maha Coucha, Deanna R. Bolduc, Veronica C. Hermanns, Stan P. Heath, Maryam Abdelghani, Lilia Z. Macias-Moriarity, Mohammed Abdelsaid

**Affiliations:** 1Department of Biomedical Sciences, School of Medicine, Mercer University, Savannah, GA 31404, USA; faith.nicole.burnett@live.mercer.edu (F.N.B.); hermanns_vc@mercer.edu (V.C.H.); stan.patrick.heath@live.mercer.edu (S.P.H.); abdelghani_m@mercer.edu (M.A.); 2Department of Pharmaceutical Sciences, School of Pharmacy, South University, Savannah, GA 31406, USA; mcoucha@southuniversity.edu (M.C.); lmacias-moriarity@southuniversity.edu (L.Z.M.-M.)

**Keywords:** SARS-CoV-2 spike protein, diabetes, hACE2 KI mice, cerebrovasculature, RAAS, TLR signaling

## Abstract

Diabetics are more vulnerable to SARS-CoV-2 neurological manifestations. The molecular mechanisms of SARS-CoV-2-induced cerebrovascular dysfunction in diabetes are unclear. We hypothesize that SARS-CoV-2 exacerbates diabetes-induced cerebrovascular oxidative stress and inflammation via activation of the destructive arm of the renin–angiotensin-aldosterone system (RAAS) and Toll-like receptor (TLR) signaling. SARS-CoV-2 spike protein was injected in humanized ACE2 transgenic knock-in mice. Cognitive functions, cerebral blood flow, cerebrovascular architecture, RAAS, and TLR signaling were used to determine the effect of SARS-CoV-2 spike protein in diabetes. Studies were mirrored in vitro using human brain microvascular endothelial cells treated with high glucose-conditioned media to mimic diabetic conditions. Spike protein exacerbated diabetes-induced cerebrovascular oxidative stress, inflammation, and endothelial cell death resulting in an increase in vascular rarefaction and diminished cerebral blood flow. SARS-CoV-2 spike protein worsened cognitive dysfunction in diabetes compared to control mice. Spike protein enhanced the destructive RAAS arm at the expense of the RAAS protective arm. In parallel, spike protein significantly exacerbated TLR signaling in diabetes, aggravating inflammation and cellular apoptosis vicious circle. Our study illustrated that SAR-CoV-2 spike protein intensified RAAS and TLR signaling in diabetes, increasing cerebrovascular damage and cognitive dysfunction.

## 1. Introduction

In late 2019, a novel severe acute respiratory syndrome coronavirus 2 (SARS-CoV-2) rapidly spread worldwide. SARS-CoV-2 viral infections caused coronavirus disease 2019 (COVID-19), which was eventually declared a global pandemic by the World Health Organization (WHO) in March 2020. COVID-19 is known chiefly for its flu-like symptoms that can progress to acute respiratory distress syndrome (ARDS) in severe cases. As of June 2023, data from the WHO state that over 760 million confirmed COVID-19 cases worldwide, with over 6 million COVID-19-related deaths [1]. COVID-19 mortality is attributed to respiratory failure, pulmonary infection, septic shock, and multiorgan failure, especially in the presence of other comorbidities such as hypertension, chronic lung disease, diabetes, and obesity [2].

With increasing vaccination rates, mortality rates associated with COVID-19 have decreased. Nonetheless, COVID-19 survivors are experiencing neurological complications. Neurological symptoms, including loss of smell and taste, depressed mood, headache, cognitive dysfunction, encephalitis, and stroke, are documented in some COVID-19 survivors [3,4]. Studies have shown that fatigue was the most reported neurological symptom of COVID-19, followed by brain fog, sleep disturbances, and memory issues [5]. Yet, the mechanisms of cerebrovascular complications are still under investigation.

Cardiovascular disorders such as obesity, diabetes, COPD, and hypertension have increased patients’ vulnerability to COVID-19 [6]. These cardiovascular comorbidities can increase the severity of COVID-19 infection and worsen patient outcomes. Recent studies showed that diabetes raises the risk of COVID-19-related mortality by up to 14% [7]. The diabetic population also suffers a higher rate of intensive care unit (ICU) admission and mechanical ventilation [7]. SARS-CoV-2 has been shown to cause severe inflammation in various organs, such as the lungs, kidneys, and brain, of diabetic patients [8,9,10,11]. Moreover, Mechi et al. reported that diabetic COVID-19 survivors experienced a greater frequency of fatigue and disruptions in concentration and memory, yet the molecular mechanisms are unclear [12]. Our study aims to investigate why diabetic populations are more vulnerable to COVID-19 cerebrovascular complications using an in vivo animal model.

The renin–angiotensin–aldosterone system (RAAS) involvement in COVID-19-induced encephalopathies is still under investigation. Angiotensin II (Ang II), the main bioactive product of the RAAS system, acts mainly through activation of the Ang II type 1 receptor (AT_1_R) and its downstream signaling cascade to induce neurovascular oxidative stress and inflammation that contributes to cognitive dysfunction. Ang(1–7), formed from Ang II by angiotensin-converting enzyme II (ACE2), counteracts Ang II/AT_1_R through its Mas receptor (MasR) and represents the protective arm of RAAS. COVID-19 spike protein (S-protein) binds and internalizes the ACE2 receptor for the invasion of viral particles into the cells, depriving RAAS of Its protective arm. A major challenge of studying SARS-CoV-2 in the laboratory is finding a suitable animal model. SARS-CoV-2 has a structural spike protein that binds to angiotensin-converting enzyme 2 (ACE2) to facilitate cellular entry. The spike protein of SARS-CoV-2 does not bind effectively with murine ACE2 [13,14]. To overcome this obstacle, we used a transgenic knock-in humanized ACE2 (hACE2) mouse as one of the acceptable COVID-19 murine models [15]. Here, we use hACE-2 to study COVID-19 cerebrovascular complications. 

This study aims to identify one of the mechanisms through which SARS-CoV-2 induces neurovascular dysfunctions that increase cerebrovascular complications and cognitive impairment in diabetes using hACE2 transgenic mice. We hypothesize that SARS-CoV-2 spike protein will increase diabetes-induced oxidative stress and inflammation via the RAAS imbalance and Toll-like receptor (TLR) signaling in the cerebrovascular environment resulting in increased neurovascular and cognitive dysfunctions. 

The results showed that SARS-CoV-2 spike protein increases diabetes-induced cerebrovascular inflammation and oxidative stress. This damage leads to increased cerebrovascular rarefaction and decreased cerebral blood flow in diabetics, exacerbating cognitive dysfunction compared to nondiabetic mice. We provide evidence that restoration of RAAS balance using losartan, an angiotensin receptor blocker, could reduce the detrimental effects of the SARS-CoV-2 spike protein in diabetes. 

## 2. Results

### 2.1. SARS-CoV-2 Spike Protein Aggravates Cerebrovascular Oxidative Stress in Diabetes

To assess the effect of SARS-CoV-2 spike protein on diabetes-induced oxidative stress, human brain microvascular endothelial cells (HBMVEC) were grown to confluency in normal or high glucose conditions and treated with SARS-CoV-2 spike protein with or without losartan. Gene expression for *NOX-5*, *Nrf-2*, and *SOD* were analyzed in whole-cell lysate using qRT-PCR. 

NOX-5 is an NADPH oxidase vascular isoform that generates reactive oxygen species. One-way ANOVA showed a significant difference between groups in *NOX-5* gene expression F(5,18) = 4.875, *p* = 0.005. Tukey post-hoc analysis revealed that treating human brain endothelial cells with spike protein significantly increased *NOX-5* expression under normal glucose (*p* = 0.02) and high glucose (*p* = 0.002) compared to control (Figure 1a).

Nuclear factor erythroid 2-related factor 2 (Nrf-2) is a transcription factor that regulates the cellular defense against oxidative insults by activating antioxidant defense genes. One-way ANOVA showed a significant difference between groups in *Nrf-2* gene expression F(5,18) = 10.313, *p* < 0.001. Tukey post-hoc analysis revealed that exposing brain microvascular endothelial cells to high glucose conditions reduced *Nrf-2* expression compared to control (*p* = 0.018). Additionally, treating the cells with spike protein significantly decreased *Nrf-2* expression under both normal (*p* = 0.001) and high glucose conditions (*p* = 0.024) compared to the control group. However, losartan restored *Nrf-2* expression (Figure 1b).

Superoxide dismutase (SOD) is an antioxidant enzyme that catalyzes the dismutation of the superoxide radical. One-way ANOVA showed a significant difference between groups in *SOD* gene expression F(5,18) = 12.740, *p* < 0.001. Tukey post-hoc analysis showed that treating the cells with spike protein significantly increased *SOD* expression under both normal (*p* < 0.001) and high glucose conditions (*p* < 0.001) compared to the control group. Losartan treatment showed higher *SOD* expression under normal glucose (*p* = 0.002)- and high glucose (*p* = 0.024)-treated spike protein compared to control (Figure 1c).

A similar trend was exhibited in vivo using diabetic hACE2 mice. Diabetes was induced using low-dose streptozotocin (STZ) followed by an eight-week high-fat diet, and mice were injected intravenously with recombinant SARS-CoV-2 spike protein. We assessed *NOX-1*, another vascular isoform of NADPH oxidase, in whole brain homogenate. One-Way ANOVA showed a significant difference between groups in *NOX-1* gene expression F(5,18) = 2.729, *p* < 0.053 (Figure 1d).

For, *Nrf-2* gene expression in whole brain homogenate, one-way ANOVA showed a significant difference between groups in *Nrf-2* gene expression F(5,18) = 6.313, *p* = 0.001. Tukey post-hoc analysis demonstrated that *Nrf-2* gene expression was significantly lower in the brains of diabetic mice compared to the control (*p* = 0.031). Furthermore, treating mice with spike protein significantly decreased *Nrf-2* expression under both normal (*p* = 0.011) and diabetic conditions (*p* = 0.001) compared to the control group. However, losartan restored *Nrf-2* gene expression only under normal conditions (*p* = 0.017) (Figure 1e).

For, *SOD* gene expression in whole brain homogenate, one-way ANOVA showed a significant difference between groups in *SOD* gene expression F(5,18) = 24.2, *p* < 0.001. Tukey post-hoc analysis demonstrated that *SOD* gene expression was significantly lower in the brains isolated from control hACE2 mice treated with spike protein (*p* = 0.008), diabetic mice (*p* = 0.001), and diabetic mice treated with spike protein (*p* = 0.001) compared to the control. Losartan did not restore *SOD* gene expression under either normal (*p* = 0.002) or diabetic conditions (*p* = 0.001) (Figure 1f).

### 2.2. SARS-CoV-2 Spike Protein Increases Inflammation in Diabetic hACE2 Mice

Next, we examined the effect of SAR-CoV-2 spike protein on diabetes-induced inflammation using qRT-PCR and Western blot analysis. HBMVEC were treated with SARS-CoV-2 under normal and high glucose conditions and expression of the inflammatory markers Il-6, Il-1β, and TNF-α were assessed. 

One-way ANOVA showed a significant difference between groups in *Il-6* gene expression F(5,18) = 50.48, *p* < 0.001. Tukey post-hoc analysis revealed that *Il-6* gene expression was significantly increased in cells treated with spike protein in the presence (*p* = 0.005) and absence of losartan (*p* = 0.001) compared to control, albeit to a different extent. High glucose conditions alone also increased *Il-6* gene expression (*p* = 0.001) compared to the control, and this increase was further enhanced by spike protein treatment (*p* = 0.001). Losartan did not reduce *Il-6* gene expression under high glucose conditions compared to the control (*p* = 0.001) (Figure 2a).

One-way ANOVA showed a significant difference between groups in *TNF-α* gene expression F(5,18) = 17.159, *p* < 0.001. Tukey post-hoc analysis showed that high glucose conditions alone increased *TNF-α* expression (*p* = 0.007) compared to the control. Furthermore, treating the cells with spike protein significantly increased *TNF-α* gene expression under both normal (*p* = 0.001) and high glucose conditions (*p* = 0.001) compared to the control group. Losartan prevented the increase in *TNF-α* gene expression only under normal conditions, high glucose treated with spike protein and losartan showed increase in *TNF-α* gene expression (*p* = 0.001) compared to control (Figure 2b).

Next, we examine the protein expression for inflammatory markers TNF-α and Il-1β. One-way ANOVA showed a significant difference between groups in TNF-α protein expression F(5,18) = 11.515, *p* < 0.001. Tukey post-hoc analysis revealed that high glucose conditions alone increased TNF-α protein expression (*p* = 0.017) compared to the control. This increase was further enhanced by spike protein treatment (*p* = 0.001). Losartan reduced TNF-α protein expression under normal and high glucose conditions treated with spike protein (Figure 2c,d).

In parallel, one-way ANOVA showed a significant difference between groups in Il-1β protein expression F(5,18) = 26.123, *p* < 0.001. Tukey post-hoc analysis demonstrated that high glucose conditions alone increased Il-1β protein expression (*p* = 0.001) compared to the control, and this increase was further enhanced by the treatment with spike protein (*p* = 0.001). However, the treatment with losartan mitigated the further increase in Il-1β protein expression under normal glucose conditions but not high glucose conditions (*p* = 0.045) treated with spike protein (Figure 2e).

We assessed inflammatory markers in diabetic hACE2 mice treated with SARS-CoV-2 spike protein. We used qRT-PCR analysis for *Il-6*, *Il-1β*, *TNF-α*, and *NFκB* expression in whole brain homogenate. Our results showed a trend similar to our in vitro studies. SARS-CoV-2 spike protein caused significant increases in all inflammatory markers in control mice, and diabetes had equivalent effects. SARS-CoV-2 spike protein intensified diabetes-induced inflammation *Il-6*, *Il-1β*, *TNF-α*, and *NFκ-B* expression in hACE2 mice. Losartan reduced inflammatory response caused by the SARS-CoV-2 spike protein. 

One-way ANOVA showed a significant difference between groups in *Il-6* gene expression F(5,18) = 20.113, *p* = 0.001. Tukey post-hoc analysis revealed that spike protein had significantly greater *Il-6* expression compared to control (*p* = 0.042), diabetes compared to control (*p* = 0.003), and diabetes treated with spike protein compared to control (*p* = 0.001) (Figure 2f).

One-way ANOVA showed a significant difference between groups in *Il-1β* gene expression F(5,18) = 6.684, *p* = 0.001. Tukey post-hoc analysis revealed that diabetes treated with spike protein had significantly greater *Il-1β* expression compared to control (*p* = 0.002) (Figure 2g).

One-way ANOVA showed a significant difference between groups in *TNF-α* gene expression F(5,18) = 9.945, *p* = 0.001. Tukey post-hoc analysis revealed that spike protein had significantly greater *TNF-α* expression compared to control (*p* = 0.006), diabetes compared to control (*p* = 0.002), and diabetes treated with spike protein compared to control (*p* = 0.005). Losartan treatment reduced *TNF-α* gene expression in control and diabetic hACE2 mice (Figure 2h).

One-way ANOVA showed a significant difference between groups in *NFκB* gene expression F(5,18) = 23.409, *p* = 0.001. Tukey post-hoc analysis revealed that diabetes had significantly greater *NFκ-B* expression compared to control (*p* = 0.001), and diabetes treated with spike protein compared to control (*p* = 0.001). Losartan treatment reduced *NFκB* gene expression in control and diabetic hACE2 mice (Figure 2i).

### 2.3. SARS-CoV-2 Spike Protein Causes Increased Vascular Rarefaction in Diabetic hACE2 Mice

We evaluated the effect of SARS-CoV-2 spike protein on endothelial cell apoptosis and vascular density. HBMVECs were treated with SARS-CoV-2 under normal and high glucose conditions. Apoptotic marker, cleaved caspase-3 protein expression levels in the whole-cell lysate were assessed using Western blot analysis. One-way ANOVA showed a significant difference between groups in cleaved caspase-3 protein expression F(5,18) = 16.325, *p* < 0.001. Tukey post-hoc analysis revealed that cleaved caspase-3 protein expression was significantly increased in cells treated with spike protein in the presence (*p* = 0.021) and absence of losartan (*p* = 0.037) compared to control. High glucose conditions alone increased cleaved caspase-3 protein expression (*p* = 0.002) compared to the control; this increase was further enhanced by treatment with spike protein (*p* = 0.001). Treating the cells with losartan prevented the increase in cleaved caspase-3 protein expression under high glucose conditions (Figure 3a,b).

In parallel, we examined the caspase-3 cleavage in hACE2 mice whole brain homogenate. One-way ANOVA showed a significant difference between groups in caspase-3 cleavage protein expression F(5,18) = 5.676, *p* = 0.003. Tukey post-hoc analysis demonstrated that cleaved caspase-3 protein expression was significantly increased only in diabetic mice treated with the spike protein (*p* = 0.002) compared to the control group (Figure 3c,d).

To further investigate vascular cell death and rarefaction in control and diabetic hACE2 brains, we assessed vascular density after SARS-CoV-2 spike protein injection. We previously showed that diabetes increases cerebrovascular pathological neovascularization [16]. We detected similar vascular density increase in diabetic hACE2 compared to control hACE2 mice. SARS-CoV-2 spike protein caused a decrease in vascular density in both control and diabetic mice, but the effects were much more prominent in the diabetic group. One-way ANOVA showed a significant difference between groups in vascular density F(5,36) = 3.322, *p* = 0.014 (Figure 3e,f). 

### 2.4. SARS-CoV-2 Spike Protein Causes Decreased Cerebral Blood Flow in Diabetic hACE2 Mice

The effect of SARS-CoV-2 spike protein on cerebral blood flow in diabetes was assessed in hACE2 mice. Diabetes was induced in hACE2 mice using low-dose STZ followed by an eight-week high-fat diet, and mice were intravenously injected with recombinant SARS-CoV-2 spike protein. Baseline superficial cerebral blood flow measurements were collected via laser speckle imaging immediately before spike protein injection. Follow-up cerebral blood flow measurements were collected every five days for a period of fifteen days following injection. Our results showed that spike protein caused a decrease in cerebral blood flow in the diabetic group that reached significance by day fifteen. One-way ANOVA showed no significant differences between groups at day zero F(5, 31) = 0.108, *p* = 0.990, and day five F(5, 31) = 1.715, *p* = 0.161. One-way ANOVA showed significant differences between groups at day ten F(5, 31) = 3.632, *p* = 0.010 and day fifteen F(5, 32)= 4.235, *p* = 0.005. Tukey post-hoc analysis revealed that diabetes treated with spike protein had significantly lower cerebral blood flow compared to control (day 15, *p* = 0.04) (Figure 4). Mice that received the spike protein experienced 34% reduction in the cerebral blood flow (70 ± 17) compared to control (106 ± 11). Although analysis did not reach statistical significance, Tukey post-hoc analysis revealed that losartan treatment significantly increased the cerebral blood flow to levels comparable to the control group (Figure 4).

### 2.5. SARS-CoV-2 Spike Protein Deteriorates Cognitive Function in Diabetic hACE2 Mice

The Y-maze was used to assess the effects of SARS-CoV-2 spike protein on the memory and learning functions of diabetics. Diabetes was induced in hACE2 mice using low-dose STZ followed by an eight-week high-fat diet. Mice were then intravenously injected with recombinant SARS-CoV-2 spike protein. All behavior testing was completed fourteen days after injection with spike protein. Time in the novel arm was used as a metric of cognitive function. One-way ANOVA showed a significant difference between groups in time spend in the novel arm F(5, 53) = 3.895, *p* = 0.004. Tukey post-hoc analysis revealed that diabetes had significantly less time in the new arm compared to control (*p* = 0.049) and diabetes treated with spike protein compared to control (*p* = 0.05). Furthermore, Tukey post-hoc analysis revealed no significance between control and other groups in the total distance traveled (Figure 5a,b). A similar trend was recorded using a Barnes maze. 

### 2.6. SARS-CoV-2 Spike Protein Disrupts the RAAS 

We assessed the effect of SARS-CoV-2 spike protein on RAAS balance and TLR signaling in diabetics. Diabetes was induced in hACE2 mice using low-dose STZ followed by an eight-week high-fat diet, and mice were intravenously injected with recombinant SARS-CoV-2 spike protein. Brain homogenate was used to assess gene expression of elements of the RAAS (*ACE2*, *Angll*, *AT_1_R*, *AT_2_R*, and *MasR*) with qRT-PCR. One-way ANOVA showed significant differences between groups in *ACE2* gene expression F(5,18) = 6.71, *p* = 0.001. Diabetes increased *ACE-2* expression. Spike protein decreased *ACE-2* expression in control and diabetic animals (Figure 6a).

One-way ANOVA showed significant differences between groups in *Ang II* gene expression F(5,18) = 6.71, *p* = 0.001. Spike protein showed a maximum increase in *Ang II* gene expression in diabetes (Figure 6b).

One-way ANOVA showed significant differences between groups in *AT_1_R* gene expression F(5,18) = 4.368, *p* = 0.009 compared to control. Tukey post-hoc analysis revealed that diabetes had significantly greater *AT_1_R* gene expression compared to control (*p* = 0.043). Moreover, diabetes treated with spike protein showed a significant increase in *AT_1_R* compared to control (*p* = 0.013) (Figure 6c).

We evaluated the protective arm of the RAAS system, mainly the *AT_2_R* and *MAS* receptor. One-way ANOVA showed significant differences between groups in *AT_2_R* gene expression F(5,17) = 3.556, *p* = 0.022 compared to control. Tukey post-hoc analysis revealed that diabetes treated with spike protein had significantly less *AT_2_R* gene expression compared to control (*p* = 0.012), which was improved with losartan treatment (Figure 6d).

Our results also showed that spike protein significantly decreases *MAS* receptor gene expression. One-way ANOVA showed significant differences between groups in *MAS* receptor gene expression F(5,18) = 12.707, *p* = 0.001 compared to control. Tukey post-hoc analysis revealed that *MAS* receptor gene expression was significantly reduced in diabetic mice compared to the control group (*p* = 0.003). Moreover, exposure to spike protein significantly reduced *MAS* receptor gene expression in control (*p* = 0.040) and diabetic mice (*p* = 0.001), which was restored with losartan treatment (Figure 6e).

Altogether, our results showed that spike protein caused an increase in gene expression of *Ang II* and *AT_1_R* and decreased the expression of the vascular protective *ACE2*, *AT_2_R*, and *MAS* receptors. 

### 2.7. SARS-CoV-2 Spike Protein Increases TLR Signaling in Diabetic hACE2 Mice

Assessment of TLR signaling began with analyzing protein expression of DAMPS (S100 and HMGB1) in hACE2 whole brain homogenate using Western blotting. SARS-CoV-2 spike protein provoked significant increases in both S100 and HMGB1 protein expression in control mice. One-way ANOVA showed significant differences between groups in S100 protein expression F(5,18) = 4.054, *p* = 0.012 compared to control. Tukey post-hoc analysis revealed that diabetes treated with spike protein had significantly greater S100 protein expression compared to control (*p* = 0.021); losartan attenuated this effect (Figure 7a,b).

One-way ANOVA showed significant differences between groups in HMGB1 protein expression F(5,18) = 2.733, *p* = 0.039 compared to control. Tukey post-hoc analysis revealed that diabetes treated with spike protein had significantly greater HMGB1 protein expression compared to control (*p* = 0.044) (Figure 7c).

Lastly, we evaluated the protein expression of TLR8, MyD88, and TRAF6 as a measure of TLR signaling. SARS-CoV-2 spike protein caused significant increases in TLR8 and MyD88 expression in control and diabetic animals. One-way ANOVA showed significant differences between groups in TLR8 protein expression F(5,18) = 4.973, *p* = 0.005 compared to control. Tukey post-hoc analysis revealed that diabetes treated with spike protein had significantly greater TLR8 protein expression compared to control (*p* = 0.004), which was attenuated with losartan (Figure 7d,e).

And for the downstream signal MyD88, one-way ANOVA showed significant differences between groups in MyD88 protein expression F(5,18) = 6.262, *p* < 0.001 compared to control. Tukey post-hoc analysis revealed that diabetes treated with spike protein had significantly greater MyD88 protein expression compared to control (*p* = 0.003), which was attenuated with losartan (Figure 7f,g).

And for the downstream signal TRAF6, one-way ANOVA showed significant differences between groups in TRAF6 protein expression F(5,18) = 9.843, *p* < 0.001 compared to control. Tukey post-hoc analysis revealed that diabetes treated with spike protein had significantly greater TRAF6 protein expression compared to control (*p* = 0.001), which was attenuated with losartan (Figure 7h,i).

## 3. Discussion

In the current study, we investigated why diabetic populations are more vulnerable to COVID-19-induced cerebrovascular complications. Our study tested the hypothesis that SARS-CoV-2 spike protein exacerbates the neurovascular complications in diabetes using an animal model. We also examined the role of the renin–angiotensin–aldosterone system (RAAS) imbalance as one of the molecular mechanisms for enhanced COVID-19-induced cerebrovascular complications in diabetes.

The main findings of our study are that (1) SARS-CoV-2 spike protein aggravates diabetes-induced cerebrovascular oxidative stress and inflammation, leading to an increase in cerebrovascular rarefaction and decreased cerebral blood flow in diabetes. (2) In addition, spike protein exacerbates cognitive dysfunction in diabetics. (3) Furthermore, losartan, an angiotensin receptor blocker (ARBs), reduced the detrimental effects of SARS-CoV-2 spike protein in the brain microvascular environment by restoring RAAS balance.

Multiple reports have confirmed neurologic manifestations of COVID-19 [17]. Aggressive neurological manifestations can predict COVID-19 severity. For example, acute encephalopathy has been linked with severe cases of COVID-19, which was attributed to COVID-19-induced hypoxic brain damage [9]. Another predictor of COVID-19 outcomes was also attributed to the presence of other cardiovascular comorbidities, such as hypertension and diabetes [18,19,20,21,22]. In particular, diabetic populations are among the top comorbidities that lead to poor outcomes of COVID-19 [23,24]. Diabetic populations faced a greater risk of developing ischemic strokes due to COVID-19 [25].

Although clinical evidence confirmed increased severity and mortality of COVID-19 patients with diabetes, the molecular mechanism is still unclear [18,19,26,27]. Our study focused on RAAS imbalance as one of the possible molecular mechanisms by which COVID-19 exacerbates diabetes-induced cerebrovascular complications.

Diabetes causes pathological microvascular changes in several organs, such as the brain, eyes, and kidneys. Our group has previously shown that diabetes causes an increase in pathological cerebral neovascularization. Our studies showed that these pathological changes in cerebrovascular microenvironments contributed to diabetes-induced cognitive dysfunction [28,29]. In the current study, we illustrated novel evidence that SARS-CoV-2 spike protein induces vascular rarefaction and decreases cerebral blood flow in diabetic animals. Our studies showed that cerebrovascular endothelial cell death were more prominent in diabetes. Moreover, our results demonstrated that the vascular rarefaction and decreased blood flow intensified cognitive dysfunction in diabetes. Our experiential results were in agreement with Mechi et al. [12]. The group reported that the diabetic COVID-19 survivors experienced a greater frequency of fatigue as well as disruptions in concentration and memory [12]. In another study, COVID-19 caused hypoperfusion in more than 80% of patients [30]. Our results were in agreement with these findings. We showed that COVID-19 decreased cerebral blood flow in hACE2 mice models. Altogether, COVID-19-induced cerebrovascular complications intensified cognitive dysfunctions [31].

Delgado-Alonso et al. reported that patients with COVID-19 showed reduced cognitive performance, especially in the attention–concentration and executive functioning, episodic memory, and visuospatial processing domains [32]. Our results confirmed these findings, where we showed a deterioration in memory and spatial learning abilities in diabetics exposed to SARS-CoV-2 spike protein. Our findings were in agreement with Lykhmus et al., who showed that fragment of SARS-CoV-2 spike protein induces neuroinflammation and impairs episodic memory of mice [33].

Multiple mechanisms have been proposed for COVID-19 neurovascular manifestation. SARS-CoV-2 has been shown to cause severe inflammation in various organs, such as the lungs and kidneys, of diabetic patients [8,11]. Increased inflammation and oxidative stress in the cerebral microenvironment is one of the most acceptable theories for COVID-19-induced cerebrovascular complications [17,34,35,36].

Our study showed that SARS-CoV-2 caused significant increases in cerebrovascular inflammation and oxidative stress. We reported an increase in brain levels of Il-6, Il-1β, TNF-α, and NFκB. In support of our findings, Rutkai et al. found that SARS-CoV-2 heightened neuroinflammation in nonhuman primates. Neuroinflammation was accompanied by increased cleaved Cas-3 expression within the CNS, indicating increased apoptosis [37]. Our findings were in agreement with these studies. We showed that SARS-CoV-2 spike protein increases endothelial cell death, as demonstrated by increases in caspase activation. In the present study, the increased inflammation caused by SARS-CoV-2 spike protein was more pronounced under diabetic conditions. The hyperglycemic conditions of diabetes have been found to lead to neuroinflammation [38]. Diabetes-induced inflammation compounded with neuroinflammation brought on by SARS-CoV-2 has been associated with increased COVID-19-related organ damage and mortality rates [39]. Our results were in agreement with Beckman et al. The group showed that SARS-CoV-2 infects neurons and induces neuroinflammation in a nonhuman primate model [36]. In our study, we illustrated that inflammation in vasculature was significantly higher in diabetic animals compared to controls. These increases in oxidative stress and inflammation in diabetes could be a reasonable explanation for the higher mortality rates seen in diabetic patients with COVID-19 compared to nondiabetic patients [40,41].

The role of RAAS in COVID-19 pathogenies and SARS-CoV-2-induced organ damage has been well established [42,43]. SARS-CoV-2 utilizes ACE2 to gain access to the interior of the cell. Diabetes causes an increase in the expression of ACE2 [44]. Our results support these findings. This could be linked to the increased virulence of SARS-CoV-2 in this vulnerable population. Our results also showed that SARS-CoV-2 spike protein downregulates the expression of *ACE2*, which has harmful consequences regarding RAAS balance. In a study conducted by Kutz et al., there was evidence that Ang l, Ang ll, and Ang1–7 peptide levels were decreased in COVID-19 patients [45], which indicates a downregulation of RAAS functions. In contrast, Carpenter et al. showed that increases in Ang ll: Ang1–7 ratios were associated with increased proinflammatory cytokine production [46]. In parallel, our study illustrated that SARS-CoV-2 caused increases in Ang ll and AT_1_R, the destructive arm of RAAS, and decreases in AT_2_R and MasR, the protective arm of RAAS. These findings are indicative of an imbalance in RAAS. Ang1–7 has been shown to reduce inflammation and oxidative stress in stroke models, providing neuroprotection [47]. Decreased expression of Ang 1–7 could shift the balance of RAAS toward AT1R activation, leading to increased vasoconstriction, fibrosis, thrombosis, oxidative stress, and inflammation [47].

The continued usage of RAAS inhibitors, including ARBs and ACE inhibitors, in treatment protocols of COVID-19-positive patients has been another source of controversy. It has been hypothesized that increases in ACE2 expression, resulting from the use of these medications, can increase the virulence of SARS-CoV-2. Several case studies have found that RAAS inhibitors are associated with decreased severity of COVID-19, lessened organ injury, and reduced inflammation [48,49]. Evidence also shows that RAAS inhibitors, including ARBs and ACE inhibitors, protect against Ang ll/Ang 1–7 imbalance [50,51]. In the current study, treatment with losartan reduced the effects of SARS-CoV-2 in the cerebral microvasculature. Mice treated with losartan showed reduced inflammation and oxidative stress following injection with spike protein. Losartan reduced the cognitive dysfunction in hACE2 mice by minimizing vascular rarefaction and changes in cerebral blood flow. This can be related to the maintenance of RAAS balance attained using losartan treatments following SARS-CoV-2 spike protein injections. These findings suggest that losartan could be a potential therapeutic option for SARS-CoV-2-induced cerebrovascular complications. [52,53]

TLR signaling is a significant contributor to the cytokine storms seen in severe cases of COVID-19. We found that protein expression of DAMPS, such as S-100, and HMGB1, were increased following exposure to spike protein. Chen et al. established that increased S100 and HMGB1 in the serum of COVID-19 patients was linked to higher mortality rates [54]. Diabetes leads to an increase in both S-100 and HMGB1 patients [55,56]. This could further explain elevated COVID-19 severity in diabetic patients. The S1 subunit of SARS-CoV-2 has also been shown to act as a pathogen-associated molecular pattern (PAMP), triggering NFκ-B and AP-1 signaling via TLR2 and TLR4 [57]. Other studies have also implicated TLR8 activation in SARS-CoV-2-induced cytokine production [58,59,60]. Our findings indicate that spike protein causes significant increases in the expression of TLR8 and downstream signaling molecules, leading to inflammation.

Our study has some limitations. In our study, we used saline as the control treatment. A better control treatment would be the use of heat-inactivated SARS-CoV-2 spike protein or inactivated virus molecule. We used only the original SARS-CoV-2 alpha variant spike protein. It would be better to detect the effect of new viral variant proteins, which have higher binding capability for the ACE2 receptor. Finally, we used male animals in most of our studies. Using female animals should be considered to detect any sex differences.

In summary, the findings of this study show that SARS-CoV-2 spike protein exacerbated diabetes-induced cerebrovascular complications. Although it was known that diabetics experience increased severity of COVID-19, there was a need for more investigation into the molecular mechanism. Our novel animal model provided evidence that the SARS-CoV-2 spike protein increases inflammation and oxidative stress in the cerebrovascular microenvironment, leading to vascular rarefaction and decreased cerebral blood flow. Diabetic mice exposed to SARS-CoV-2 spike protein had significant increases in cognitive dysfunction due to the worsening of preexisting damage to the cerebral vasculature. We demonstrated signs of increased RAAS imbalance and TLR signaling following exposure to SARS-CoV-2 spike protein, allowing us to elucidate how spike protein induces cerebrovascular complications. We also addressed the debate of whether ARBs had any beneficial effect on COVID-19 patients. Mice treated with losartan following spike protein injection had significantly better outcomes, indicating that ARBs could be a potential therapeutic option for SARS-CoV-2-induced cerebrovascular complications.

## 4. Materials and Methods

### 4.1. Animals

B6(Cg)-*Ace2^tm1.1(ACE2)Mdk^*/J mice (Jax lab: Stock No: 035800), also known as humanized ACE2 knock-in (hACE2 KI) mice, were purchased from Jackson Laboratory (Ellsworth, ME, USA). Mice were inbred at the animal facility of Mercer University. Animal studies were conducted according to the ethical guidelines of the National Institutes of Health’s *Guide for the Care and Use of Laboratory Animals* and adhered to ARRIVE guidelines 2.0. All animal protocols were approved and monitored by the Mercer University Institutional Animal Care and Use Committee (IACUC), which is accredited by the American Association for Accreditation of Laboratory Animal Care. Animals maintained a standard rat chow diet and were provided unlimited tap water access. Mice were kept on a 12 h light–dark cycle.

### 4.2. Induction of Diabetes and Treatment

Eight-week-old male hACE2 KI mice were randomly assigned to control or diabetic groups and fasted overnight. Control mice received an intraperitoneal vehicle injection and continued a standard rat chow diet. Mice in the diabetic group received an intraperitoneal injection of low-dose streptozotocin (35 mg/kg body weight, Adipogen-Life Science, San Diego, CA, USA, Cat AG-CN2-0046) and were switched to a high-fat diet (HFD, 45 kcal% fat, Research Diets Inc., New Brunswick, NJ, USA, Cat. D12492) for eight weeks following STZ injection to induce type 2 diabetes. The body weight and blood glucose of the animals were recorded weekly. Successful diabetes induction was confirmed with a glucose tolerance test conducted seven weeks after induction of diabetes (Appendix A). Control mice weighed 25.47 ± 1.52 g on week eight of the experiment. Diabetic mice weighed 36 ± 3.93 g. Spike protein and losartan treatment did not cause any significant changes in mice body weight. Control mice had a blood glucose of 201.8 ± 12.2 mg/dL while mice in the diabetic group had a blood glucose of 239.3 ± 36.1 mg/dL. Control mice injected with spike protein had an average blood glucose of 243.7 ± 61.6 mg/dL one week after injection. Diabetic mice injected with spike protein had an average blood glucose of 251 ± 35.2 mg/dL. Losartan treatment did not have any effect on blood glucose levels.

Eight weeks after diabetes induction, control and diabetic mice were randomly assigned to six groups: (1) control, (2) diabetic, (3) control + SARS-CoV-2 spike protein injection, (4) diabetic + SARS-CoV-2 spike protein injection, (5) control + SARS-CoV-2 spike protein injection + losartan, and (6) diabetic + SARS-CoV-2 spike protein injection + losartan. Spike protein was injected intravenously via the jugular vein with SARS-CoV-2 nucleoprotein/spike protein recombinant protein (4 µg/animal, Invitrogen, Rockford, IL, USA, Cat. No. RP-87706). Losartan (10 mg/kg body weight, Tokyo Chemical Industry, Tokyo, Japan, Cat. No. L0232) was added to the water supply of the animal cages. Treatment with losartan began immediately after SARS-CoV-2 recombinant spike protein injection.

### 4.3. Assessment of Cognitive Functioning

Memory and learning functions were assessed at baseline and ten days after SARS-CoV-2 recombinant spike protein injection using Y-shape maze.

### 4.4. Y-Maze

The Y-maze from Stoelting Co. (Wood Dale, IL, USA) is a maze composed of three arms in the shape of a Y, used to test spatial learning and memory. During training, animals were placed into the Y-maze with one arm (the novel arm) blocked off. On test day, the animals were granted access to all three arms of the maze. Movement around the maze was measured and recorded using ANYmaze software (Version 6.1). The ratio of time spent in novel arm to all arms and the total distance traveled were collected from ANYmaze software and analyzed blindly.

### 4.5. Assessment of Cerebral Blood Flow

Cerebral blood flow was measured with the RFLSI Ⅲ Laser Speckle Imaging System (RWD, San Diego, CA, USA). Baseline readings were collected prior to injections with SARS-CoV-2 spike protein, and subsequent readings were collected every five days following injection for fifteen days. Mice were anesthetized using isoflurane and a vertical incision was made to expose the mouse’s skull. LSCI software (Version V5.0) was used, set to the RWD laser speckle imaging setting, to record the cerebral blood flow. The surgical area was cleaned and closed with a clip. Perfusion per area on days 5, 10, and 15 were compared as a percentage of the baseline readings of the control group.

### 4.6. Vascular Density Assessment

hACE2 mice brains were collected, fixed in 2% paraformaldehyde, and dehydrated in 30% sucrose. Brains were sectioned at 25–30 µm thickness using a Leica CM3050 S cryostat (Leica Biosystems Inc. Buffalo Grove, IL, USA). Brain sections were stained with Lycopersicon esculentum lectin, DyLight™ 488 (Vector Laboratories, Burlingame, CA, USA, Cat. No. DL-1174-1). Sections were imaged using a Nikon Eclipse Ti-E Inverted Microscope (Nikon Instruments Inc., Melville, NY) in the program NiS Elements AR. FIJI software (version V1.54g) was used to analyze the 3D constructs of the Z-stacked images, and vascular density was calculated. Vascular density was calculated by dividing the mean density of stained vasculature, as determined by FIJI, by the total number of planes in the Z-stack.

### 4.7. Cell Culture

Primary human brain microvascular endothelial cells (HBMVECs) were purchased from Angio-Proteomie (Boston, MA, USA, Cat. No. cAP-0002). Cells were grown in complete media (MCDB-131 Complete (VEC Technologies, Inc., Rensselaer, NY, USA)). All experiments were completed in duplicates using passage 6–12. HBMVECs were either treated with D-glucose (25 mM) and sodium palmitate (200 µM) to mimic the high glucose conditions of diabetes or remained in complete media with equimolar L-glucose, an osmotic control, for normal glucose condition. Cells were treated with SARS-CoV-2 recombinant spike protein (100 µM) and/or losartan (100 µM) for 48 h.

### 4.8. Polymerase Chain Reaction

Triazole (Thermo-Fisher, Waltham, MA, USA, Cat. No. AC345480250) was used to isolate RNA from brain and cellular lysate. The Thermo Scientific NanoDrop 2000C Spectrophotometer (Thermo Scientific, Waltham, MA, USA) was used to quantify RNA concentrations. The QuantStudio™ 3 Real-Time PCR System (Applied Biosystems, Thermo Scientific, Waltham, MA, USA) was utilized to run qRT-PCR. All primer sequences used are detailed in Appendix A. GAPDH was used for normalization in all experiments.

### 4.9. Immunoblotting

HBMVEC and hACE2 brain tissues were homogenized in RIPA buffer (Millipore, Billerica, MA, USA, Cat# 3P 20188). After equalizing protein loads, electrophoresis was used to separate samples on a 10% SDS–polyacrylamide gel using the Mini PROTEAN Tetra Cell SDS-PAGE Gel electrophoresis kit (Biorad Laboratories Inc., Hercules, CA, USA). Gels were transferred onto nitrocellulose membranes using the Bio-Rad Trans-Blot Turbo (Biorad Laboratories Inc., Hercules, CA, USA). Membranes were blocked and incubated in primary antibodies (1:500) overnight. Membranes were then incubated with an appropriate horseradish peroxidase-conjugated secondary antibody (1:5000) for two hours. The Azure Biosystems c600 (Azure Biosystems Inc., Dublin, CA, USA) was used to image membranes treated with Western chemiluminescent HRP Substrate (Millipore, Burlington, MA, USA). Image-J software (Version v1.50i) was used to quantify band intensity. β-actin was used for normalization. All antibodies used are listed in Appendix A.

### 4.10. Statistical Analysis

Data analysis was conducted using SPSS version 28 for all analyses. For animal studies, the sample size was determined from our previous work. One-way ANOVA was used to assess the differences in the means between control, diabetics, and S-protein with/without losartan treatment. In vitro cell culture studies were analyzed using one-way ANOVA. Significance was determined at *p* < 0.05. Data are presented as mean ± standard deviation. A Tukey’s post-hoc test was used to adjust for multiple comparisons to assess significant interaction effects from all analyses. Shapiro–Wilk formal test for normality was employed to assess the data distribution.

## Figures and Tables

**Figure 1 ijms-24-16394-f001:**
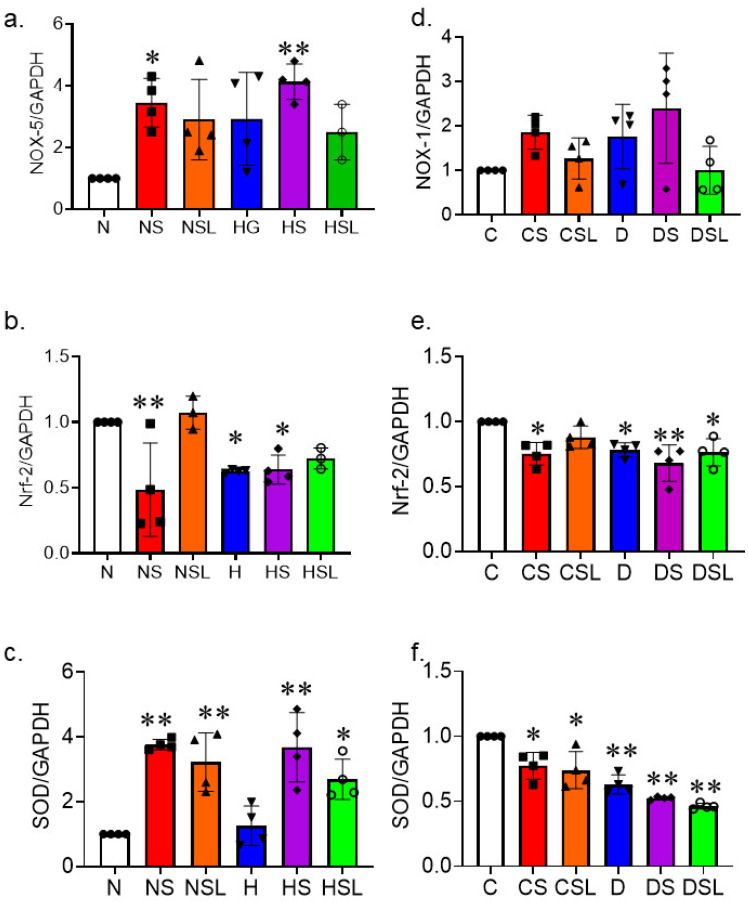
SARS-CoV-2 spike protein exacerbated diabetes-induced oxidative stress. Human brain microvascular endothelial cells were grown to confluency in complete media. Cells were then treated with SARS-CoV-2 spike protein and losartan under either normal glucose (complete media with equimolar L-glucose) or high glucose (D-glucose (25 mM) and sodium palmitate (200 µM)) conditions. (N, closed circles) normal glucose, (NS, closed squares) normal glucose + spike protein, (NSL, closed triangles) normal glucose + spike protein + losartan, (H, inverted triangles) high glucose, (HS, closed diamonds) high glucose + spike protein, (HSL, open circles) high glucose + spike protein + losartan. RNA was isolated and used for qRT-PCR assessment of oxidative stress. (**a**) RT-PCR analysis results show that spike protein exacerbated increases in *NOX-5* caused by high glucose conditions. Losartan reduced the increases in *NOX-5* gene expression following S-protein exposure under high glucose conditions (one-way ANOVA, * *p* = 0.02, ** *p* = 0.002, *n* = 4). (**b**) RT-PCR analysis results show that spike protein decreases cellular antioxidant defense by reducing *Nrf-2* gene expression (one-way ANOVA, ** *p* = 0.001, * *p* = 0.018, *n* = 4). (**c**) RT-PCR analysis results show that spike protein decreases *SOD* gene expression (one-way ANOVA, ** *p* < 0.001, ** *p* = 0.002, *n* = 4). Type 2 diabetes was induced in hACE2 mice with low-dose STZ (35 mg/kg), followed by a high-fat diet (45 kcal% fat) for eight weeks. Mice were intravenously injected with SARS-CoV-2 spike protein (4 µg/animal) and treated with losartan (10 mg/kg body weight). (C, closed circles) control hACE2, (CS, closed squares) hACE2 + spike protein, (CSL, closed triangles) hACE2 + spike protein + losartan, (D, inverted triangles) diabetic hACE2, (DS, closed diamond) diabetic hACE2 + spike protein, (DSL, open circles) diabetic hACE2 + spike protein + losartan. RNA was isolated from whole brain lysate and used for qRT-PCR assessment of oxidative stress. (**d**) RT-PCR analysis results for *NOX-1* gene expression (one-way ANOVA, *p* < 0.053, *n* = 4). (**e**) RT-PCR analysis for *Nrf-2 gene* expression in hACE2 mice (one-way ANOVA, * *p* = 0.011, ** *p* = 0.001, *n* = 4). (**f**) RT-PCR analysis for *SOD* expression in hACE2 mice (one-way ANOVA, * *p* = 0.008, ** *p* = 0.001, *n* = 4).

**Figure 2 ijms-24-16394-f002:**
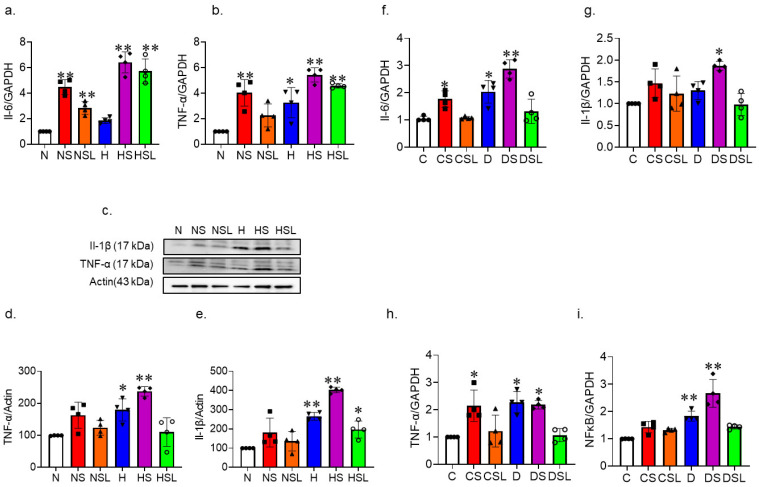
SARS-CoV-2 spike protein aggravates diabetes-induced inflammation. Human brain microvascular endothelial cells were grown to confluency in complete media. Cells were then treated with SARS-CoV-2 spike protein and losartan under either normal glucose (complete media with equimolar L-glucose) or high glucose (D-glucose (25 mM) and sodium palmitate (200 µM)) conditions. (N, closed circles) normal glucose, (NS, closed squares) normal glucose + spike protein, (NSL, closed triangles) normal glucose + spike protein + losartan, (H, inverted triangles) high glucose, (HS, closed diamonds) high glucose + spike protein, (HSL, open circles) high glucose + spike protein + losartan. RNA and protein were isolated from cell lysate and used for qRT-PCR and Western blot assessment of inflammation. (**a**) RT-PCR analysis showing the effect of spike protein on *Il-6* gene expression (one-way ANOVA, ** *p* = 0.001, *n* = 4). (**b**) RT-PCR analysis showing effect of spike protein on *TNF-α* gene expression (one-way ANOVA, * *p* = 0.007, ** *p* = 0.001, *n* = 4). (**c**) Western blot representative of TNF-α and Il-1β. (**d**) Western blot analysis for TNF-α (one-way ANOVA, * *p* = 0.017, ** *p* = 0.001, *n* = 4). (**e**) Western blot analysis for Il-1β (one-way ANOVA, ** *p* = 0.001, * *p* = 0.045, *n* = 4). Type 2 diabetes was induced in hACE2 mice with low-dose STZ (35 mg/kg), followed by a high-fat diet (45 kcal% fat) for eight weeks. Mice were intravenously injected with SARS-CoV-2 spike protein (4 µg/animal) and treated with losartan (10 mg/kg body weight). (C, closed circles) control hACE2, (CS, closed squares) hACE2 + spike protein, (CSL, closed triangles) hACE2 + spike protein + losartan, (D, inverted triangles) diabetic hACE2, (DS, closed diamonds) diabetic hACE2 + spike protein, (DSL, open circles) diabetic hACE2 + spike protein + losartan. RNA was isolated from whole brain lysate and used for qRT-PCR assessment of inflammatory markers. We used qRT-PCR analysis for *Il-6*, *Il-1β*, *TNF-α*, and *NFκB* expression in whole brain homogenate. (**f**) RT-PCR analysis for *Il-6* gene expression (one-way ANOVA, * *p* = 0.042, ** *p* = 0.001, *n* = 4). (**g**) RT-PCR analysis for *Il-1β* gene expression (one-way ANOVA, * *p* = 0.002, *n* = 4). (**h**) RT-PCR analysis for *TNF-α* gene expression (one-way ANOVA, * *p* = 0.006, *n* = 4). (**i**) RT-PCR analysis for *NFκB* gene expression (one-way ANOVA, ** *p* = 0.001, *n* = 4).

**Figure 3 ijms-24-16394-f003:**
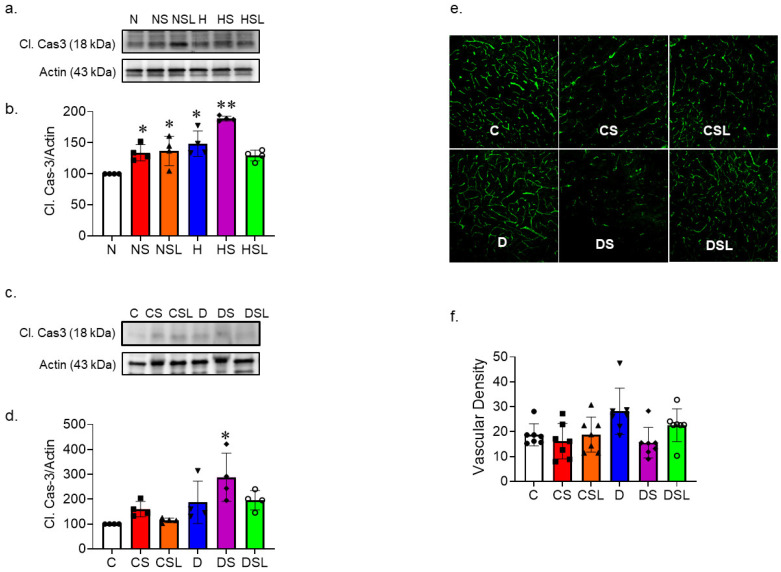
SARS-CoV-2 spike protein increases diabetes-induced endothelial cell death and vascular rarefaction. Human brain microvascular endothelial cells were grown to confluency in complete media. Cells were then treated with SARS-CoV-2 spike protein and losartan under either normal glucose (complete media with equimolar L-glucose) or high glucose (D-glucose (25 mM) and sodium palmitate (200 µM)) conditions. (N, closed circles) normal glucose, (NS, closed squares) normal glucose + spike protein, (NSL, closed triangles) normal glucose + spike protein + losartan, (H, inverted triangles) high glucose, (HS, closed diamonds) high glucose + spike protein, (HSL, open circles) high glucose + spike protein + losartan. Cell lysate was used for Western blot for assessment of endothelial cell death. (**a**,**b**) Western blot representative and quantification for cleaved caspase-3 protein expression (One-way ANOVA, * *p* = 0.021, ** *p* = 0.001, *n* = 4). (**c**,**d**) Type 2 diabetes was induced in hACE2 mice with low-dose STZ (35 mg/kg), followed by a high-fat diet (45 kcal% fat) for eight weeks. Mice were intravenously injected with SARS-CoV-2 spike protein (4 µg/animal) and treated with losartan (10 mg/kg body weight). (C, closed circles) control hACE2, (CS, closed squares) hACE2 + spike protein, (CSL, closed triangles) hACE2 + spike protein + losartan, (D, inverted triangles) diabetic hACE2, (DS, closed diamonds) diabetic hACE2 + spike protein, (DSL, open circles) diabetic hACE2 + spike protein + losartan. Whole brain homogenate was used to assess apoptosis using immunoblotting (one-way ANOVA, * *p* = 0.002, *n* = 4). (**e**,**f**) Vascular cell death and rarefaction was assessed in brain using immunohistochemical staining. Brains were sectioned and stained with Lycopersicon esculentum lectin, DyLight™ 488 to analyze cerebrovascular architecture. Diabetes alone increased vascular density compared to control hACE2 mice. SARS-CoV-2 spike protein caused a decrease in vascular density in both control and diabetic mice, but the effects were much more prominent in the diabetic group (20× magnification, one-way ANOVA, *p* = 0.014, *n* = 6–7).

**Figure 4 ijms-24-16394-f004:**
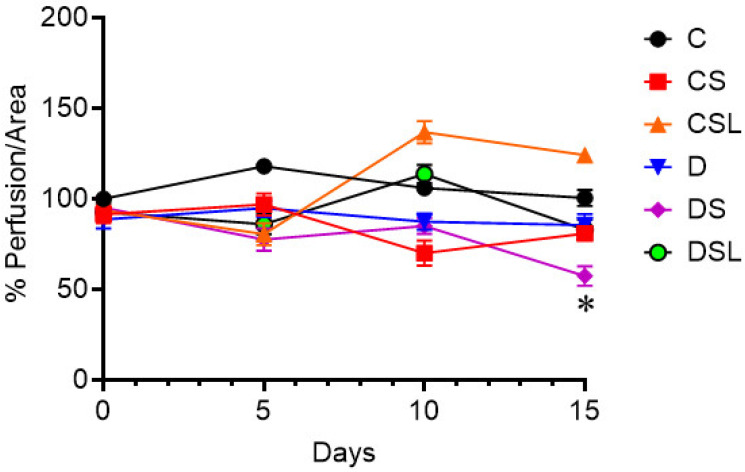
SARS-CoV-2 spike protein exacerbate cerebral blood flow in diabetics. Type 2 diabetes was induced in hACE2 mice with low-dose STZ (35 mg/kg), followed by a high-fat diet (45 kcal% fat) for eight weeks. Mice were intravenously injected with SARS-CoV-2 spike protein (4 µg/animal) and treated with losartan (10 mg/kg body weight). (C) control hACE2, (CS) hACE2 + spike protein, (CSL) hACE2 + spike protein + losartan, (D) diabetic hACE2, (DS) diabetic hACE2 + spike protein, (DSL) diabetic hACE2 + spike protein + losartan. Laser speckle imaging was used to measure cerebral blood flow for a 15-day period following S-protein injection (one-way ANOVA, day 15, * *p* = 0.04, *n* = 5–6).

**Figure 5 ijms-24-16394-f005:**
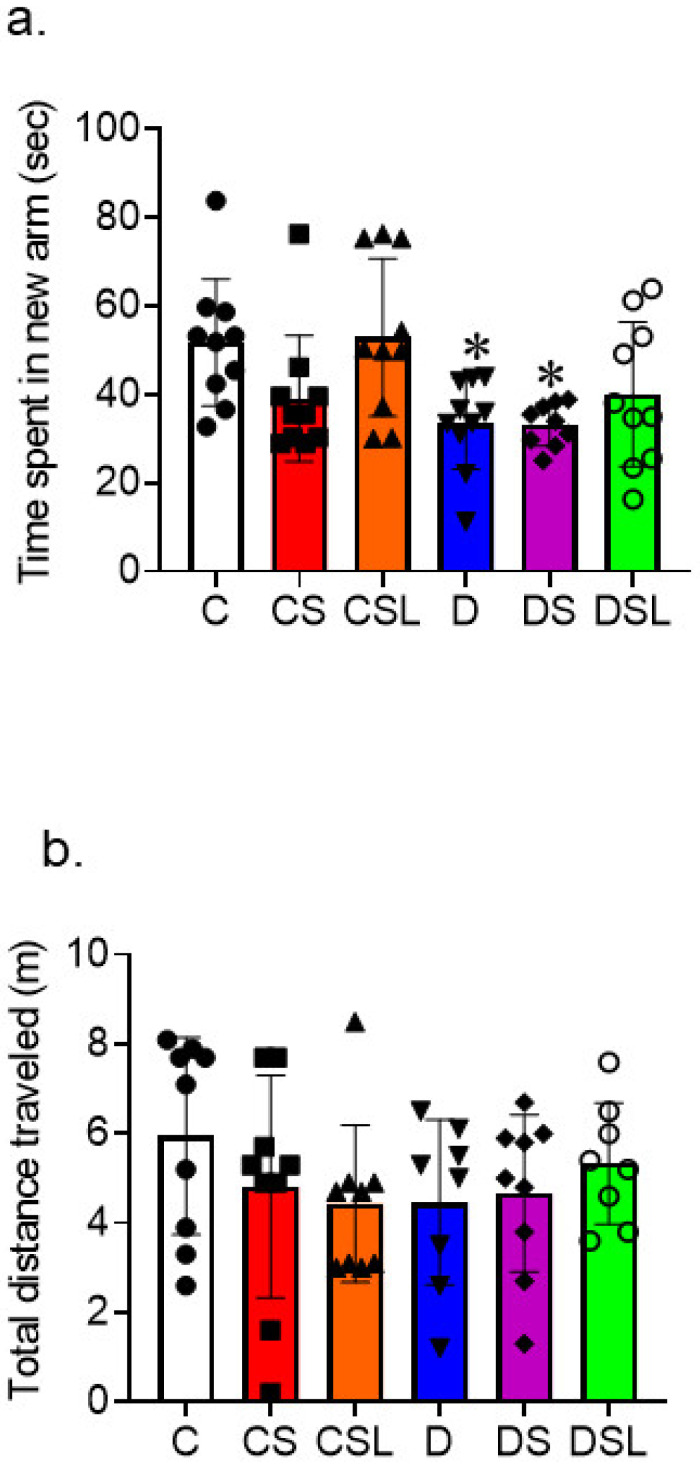
SARS-CoV-2 spike protein exacerbates diabetes-induced vascular contribution to cognitive impairment and dementia (VCID) in hACE2 mice. Type 2 diabetes was induced in hACE2 mice with low-dose STZ (35 mg/kg), followed by a high-fat diet (45 kcal% fat) for eight weeks. Mice were intravenously injected with SARS-CoV-2 spike protein (4 µg/animal) and treated with losartan (10 mg/kg body weight). (C, closed circles) control hACE2, (CS, closed squares) hACE2 + spike protein, (CSL, closed triangles) hACE2 + spike protein + losartan, (D, inverted triangles) diabetic hACE2, (DS, closed diamonds) diabetic hACE2 + spike protein, (DSL, open circles) diabetic hACE2 + spike protein + losartan. The Y-maze was used to assess learning and spatial memory 14 days after spike protein injection. (**a**) Time spent in the novel arm was used as a measure of cognitive function (one-way ANOVA, * *p* = 0.05, *n* = 8–10). (**b**) There were no significant changes in total distance travelled between groups. (*n* = 8–10).

**Figure 6 ijms-24-16394-f006:**
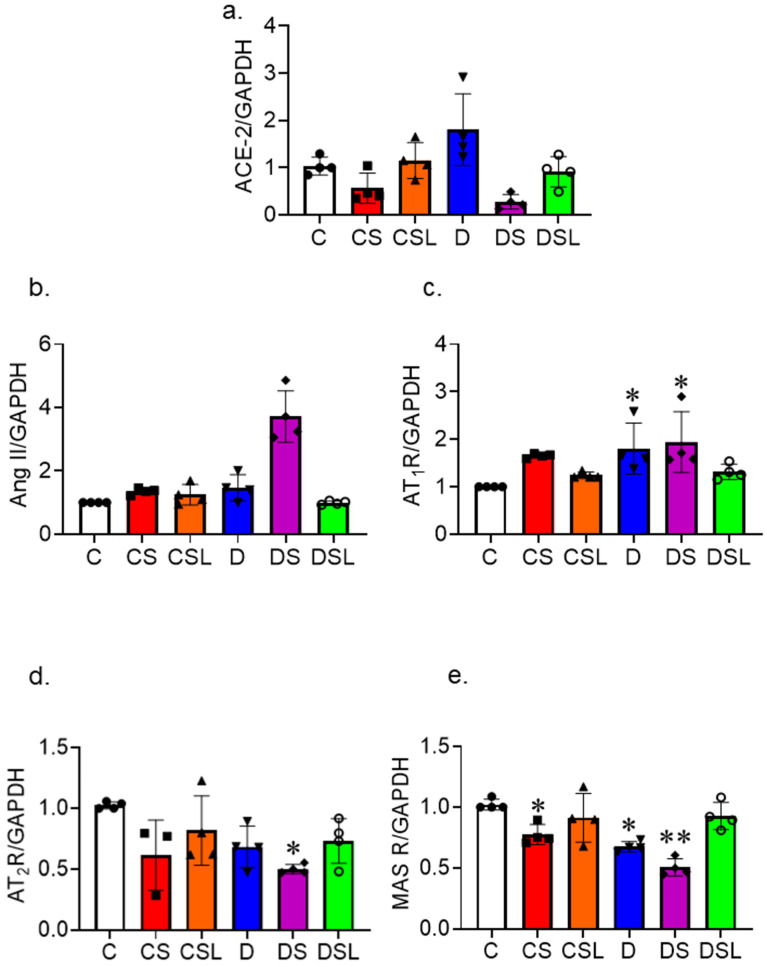
SARS-CoV-2 spike protein disrupts RAAS balance in diabetes. Type 2 diabetes was induced in hACE2 mice with low-dose STZ (35 mg/kg), followed by a high-fat diet (45 kcal% fat) for eight weeks. Mice were intravenously injected with SARS-CoV-2 spike protein (4 µg/animal) and treated with losartan (10 mg/kg body weight). (C, closed circles) control hACE2, (CS, closed squares) hACE2 + spike protein, (CSL, closed triangles) hACE2 + spike protein + losartan, (D, inverted triangles) diabetic hACE2, (DS, closed diamonds) diabetic hACE2 + spike protein, (DSL, open circles) diabetic hACE2 + spike protein + losartan. We assessed the effect of SARS-CoV-2 spike protein on RAAS balance in diabetics using RT-PCR and Western blot. (**a**) RT-PCR analysis for *ACE2* gene expression in brain homogenate (one-way ANOVA, *p* = 0.001, *n* = 4). (**b**) RT-PCR analysis for *Ang II* gene expression in whole brain homogenate (one-way ANOVA, *p* = 0.001, *n* = 4). (**c**) RT-PCR analysis for *AT_1_R* gene expression in whole brain homogenate (one-way ANOVA, * *p* = 0.043, *n* = 4). (**d**) RT-PCR analysis for *AT_2_R* gene expression in whole brain homogenate (one-way ANOVA, * *p* = 0.012, *n* = 4. (**e**) RT-PCR analysis for *MAS* receptor gene expression in whole brain homogenate (one-way ANOVA, * *p* = 0.003, ** *p* = 0.001, *n* = 4).

**Figure 7 ijms-24-16394-f007:**
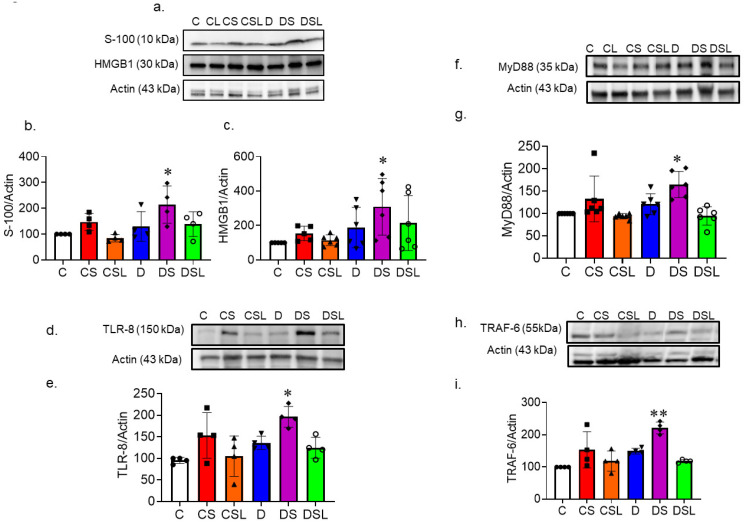
SARS-CoV-2 spike protein increases TLR signaling in diabetes. Type 2 diabetes was induced in hACE2 mice with low-dose STZ (35 mg/kg), followed by a high-fat diet (45 kcal% fat) for eight weeks. Mice were intravenously injected with SARS-CoV-2 spike protein (4 µg/animal) and treated with losartan (10 mg/kg body weight). (C, closed circles) control hACE2, (CS, closed squares) hACE2 + spike protein, (CSL, closed triangles) hACE2 + spike protein + losartan, (D, inverted triangles) diabetic hACE2, (DS, closed diamonds) diabetic hACE2 + spike protein, (DSL, open circles) diabetic hACE2 + spike protein + losartan. We assessed the effect of SARS-CoV-2 spike protein on TLR signaling in diabetics using RT-PCR and Western blot. (**a**) Western blot representative for S100 and HMGB1protein expression in whole brain homogenate. (**b**) Western blot quantification for S100 (one-way ANOVA, * *p*= 0.021, *n* = 4). (**c**) Western blot quantification for HMGB1 (one-way ANOVA, * *p* = 0.044, *n* = 4). (**d**,**e**) Western blot representative and quantification for TLR-8 protein expression in whole brain homogenate (one-way ANOVA, * *p* = 0.004, *n* = 4). (**f**,**g**) Western blot representative and quantification for MyD88 protein expression in whole brain homogenate (one-way ANOVA, * *p* = 0.003, *n* = 4). (**h**,**i**) Western blot representative and quantification for TRAF6 protein expression in whole brain homogenate (one-way ANOVA, ** *p* = 0.001, *n* = 4).

## Data Availability

Data available upon request.

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
