# Peer review of "SARS-CoV-2 Spike Protein Intensifies Cerebrovascular Complications in Diabetic hACE2 Mice through RAAS and TLR Signaling Activation"

_ijms, 2023, doi:10.3390/ijms242216394_

Round 1
Reviewer 1 Report
Comments and Suggestions for Authors
1: The study focuses specifically on the spike protein of SARS-CoV-2. While the spike protein is essential for viral entry and interaction with host cells, COVID-19 is a complex disease involving multiple viral proteins and host responses. The effects of the spike protein alone may not fully represent the effects of the entire virus. Also, spike protein administration was not based on body weight, and there was a big difference in body weight between diabetic and non-diabetic mice.
2: This manuscript states that primary human brain microvascular endothelial cells (HBMVEC) were used to mimic the high glucose conditions of diabetes using D-glucose (25mM) and sodium palmitate (200 µM). This in vitro model effectively serves as a hybrid model of glucotoxicity and lipotoxicity. However, this study did not address lipotoxicity.
3: Spike protein had no significant effect on non-fasting blood glucose. Did it have any effect on fasting blood glucose?
4: Information about antibodies and reagents is required in the Materials and Methods section.
Comments on the Quality of English LanguageNo
Author Response
Comments to Reviewer 1
First, we would like to thank Reviewer 1 for his/her valuable insights into our study. We believe the revised manuscript has improved substantially after addressing all the reviewer’s concerns. Below is our point-to-point comment on their concern.
1: The study focuses specifically on the spike protein of SARS-CoV-2. While the spike protein is essential for viral entry and interaction with host cells, COVID-19 is a complex disease involving multiple viral proteins and host responses. The effects of the spike protein alone may not fully represent the effects of the entire virus. Also, spike protein administration was not based on body weight, and there was a big difference in body weight between diabetic and non-diabetic mice.
We totally agree with the reviewer that the injection of spike protein would have a lesser damaging effect than using the whole SARS-CoV-2 viral particles. We have included this as a study limitation in the discussion section. It is worth mentioning that the spike protein used in our study is not solely a SARS-CoV-2 spike protein but rather a mixture of recombinant SARS-CoV2 nucleocapsid protein and spike protein ( Invitrogen, Thermo-Fisher, Catalog # RP-87706, USA).
Moreover, multiple research labs have confirmed that the cardiovascular damaging effect of the SARS-CoV-2 virus occurs mainly through the deprivation of the RAAS system of its angiotensin-converting enzyme 2 (ACE2) beneficial effect [1-4]. The spike protein binds strongly to human ACE2 receptors on target cells. Moreover, the spike protein plays the most important role in viral attachment, fusion, and cell entry. Working with live or attenuated viruses would require higher biosafety regulations that are not present in many research facilities. Our study will also provide evidence of the effect of SARS-CoV-2 spike protein on normal subjects, which most of the COVID-19 mRNA vaccines generate in our bodies without the entire viral particle.
We agree with the reviewer that the dose of spike protein was the same for control and diabetic animals (4 ug/animal), although diabetic animals weigh more. The reason that we used the dose is that this dose was used before by other labs to induce respiratory inflammation [5]. Second, we standardized the dose for both the control and diabetic to ensure that any detected deteriorating effects were due to the diabetic condition rather than a dose disparity.
2: This manuscript states that primary human brain microvascular endothelial cells (HBMVEC) were used to mimic the high glucose conditions of diabetes using D-glucose (25mM) and sodium palmitate (200 µM). This in vitro model effectively serves as a hybrid model of glucotoxicity and lipotoxicity. However, this study did not address lipotoxicity.
We totally agree with the reviewer about his/her concern that adding sodium palmitate to our endothelial cell culture could cause lipotoxicity. We and others had used this model as an in vitro cell culture model to mimic obese and diabetic conditions [6-8]. We did not encounter any lipotoxicity in our cell culture model at the used concentration (200 µM).
3: Spike protein had no significant effect on non-fasting blood glucose. Did it have any effect on fasting blood glucose?
We thank the reviewer for this important question. As we mentioned in the method sections, our blood glucose measuring for the glucose tolerance test was the only glucose measuring under fasting conditions. The glucose tolerance test was tested before spike protein injection. All weekly measurements after spike protein injection were under non-fasting conditions. Our results showed that spike protein caused a slight increase in blood glucose measurements in control and diabetic animals, which did not reach significance when compared to control un-injected animals.
Recent clinical reviews reported that COVID-19 infections could increase blood glucose levels and worsen diabetes complications in diabetic patients. COVID-19 increased vascular inflammation, and elevating stress conditions in the body could increase endogenous anti-insulin steroids. Moreover, pancreatic cells express ACE2, making them good targets for SARS-CoV-2, which can affect pancreatic insulin secretion, which will be accompanied by poor glycemic control. [9-12] To study these findings would be ideal for a separate study that is out of the scope of the present study.
4: Information about antibodies and reagents is required in the Materials and Methods section.
The manuscript has been updated as directed. All antibodies, reagents, and PCR primers are included in the manuscript. Please see the supplementary materials.
- Al Jabi MS, Akram Z, Bolduc D, Coucha M, Abdelsaid M. Abstract 53: Covid-19 Spike-protein Causes Cerebrovascular Rarefaction And Deteriorates Cognitive Functions In A Mouse Model Of Humanized ACE2. Stroke 2022; 53(Suppl_1):A53-A53.
- Aleksova A, Ferro F, Gagno G, Cappelletto C, Santon D, Rossi M, et al. COVID-19 and renin-angiotensin system inhibition: role of angiotensin converting enzyme 2 (ACE2) - Is there any scientific evidence for controversy? J Intern Med 2020; 288(4):410-421.
- Angeli F, Zappa M, Reboldi G, Trapasso M, Cavallini C, Spanevello A, et al. The pivotal link between ACE2 deficiency and SARS-CoV-2 infection: One year later. Eur J Intern Med 2021; 93:28-34.
- Busse LW, Chow JH, McCurdy MT, Khanna AK. COVID-19 and the RAAS-a potential role for angiotensin II? Crit Care 2020; 24(1):136.
- Johansen MD, Irving A, Montagutelli X, Tate MD, Rudloff I, Nold MF, et al. Animal and translational models of SARS-CoV-2 infection and COVID-19. Mucosal Immunol 2020; 13(6):877-891.
- Abdelsaid M, Prakash R, Li W, Coucha M, Hafez S, Johnson MH, et al. Metformin treatment in the period after stroke prevents nitrative stress and restores angiogenic signaling in the brain in diabetes. Diabetes 2015; 64(5):1804-1817.
- Abdelsaid M, Williams R, Hardigan T, Ergul A. Linagliptin attenuates diabetes-induced cerebral pathological neovascularization in a blood glucose-independent manner: Potential role of ET-1. Life Sci 2016; 159:83-89.
- Bailey J, Coucha M, Bolduc DR, Burnett FN, Barrett AC, Ghaly M, et al. GLP-1 receptor nitration contributes to loss of brain pericyte function in a mouse model of diabetes. Diabetologia 2022.
- Abdoli S, Silveira M, Doosti-Irani M, Fanti P, Miller-Bains K, Pavin EJ, et al. Cross-national comparison of psychosocial well-being and diabetes outcomes in adults with type 1 diabetes during the COVID-19 pandemic in US, Brazil, and Iran. Diabetol Metab Syndr 2021; 13(1):63.
- Buicu AL, Cernea S, Benedek I, Buicu CF, Benedek T. Systemic Inflammation and COVID-19 Mortality in Patients with Major Noncommunicable Diseases: Chronic Coronary Syndromes, Diabetes and Obesity. J Clin Med 2021; 10(8).
- Ejaz H, Alsrhani A, Zafar A, Javed H, Junaid K, Abdalla AE, et al. COVID-19 and comorbidities: Deleterious impact on infected patients. J Infect Public Health 2020; 13(12):1833-1839.
- Flaherty GT, Hession P, Liew CH, Lim BCW, Leong TK, Lim V, et al. COVID-19 in adult patients with pre-existing chronic cardiac, respiratory and metabolic disease: a critical literature review with clinical recommendations. Trop Dis Travel Med Vaccines 2020; 6:16.
Reviewer 2 Report
Comments and Suggestions for Authors
This study explores how the SARS-CoV-2 spike protein exacerbates cerebrovascular complications in diabetic individuals, leading to oxidative stress, inflammation, and cognitive dysfunction. The study also reveals an increased imbalance in the Renin-Angiotensin-Aldosterone System (RAAS) and Toll-like receptor (TLR) signaling following spike protein exposure in diabetic mice models. It proposes that angiotensin receptor blockers, such as Losartan, could be a potential treatment to restore balance in the RAAS and mitigate these complications, offering a potential therapeutic option for managing cerebrovascular complications induced by SARS-CoV-2.
Overall, the manuscript is well-written, and the results support the proposed hypothesis. However, I have the following comments to improve the manuscript:
1. The authors' claim of restoring RAAS balance using Losartan, an angiotensin receptor blocker, is not novel (line 81). Angiotensin II AT1 receptor blockers as treatments for inflammatory brain disorders have already been explored.
(Saavedra JM. Angiotensin II AT(1) receptor blockers as treatments for inflammatory brain disorders. Clin Sci (Lond). 2012 Nov;123(10):567-90. doi: 10.1042/CS20120078. PMID: 22827472; PMCID: PMC3501743.)
2. The authors should have justified their claim regarding how their mice model is a novel animal model.
3. The results section is unnecessarily lengthy. Figure captions can be shortened by removing repeated statements that are already discussed in the main text. Combine the statements discussing results obtained for groups of genes expressed in response to specific molecular signaling.
4. The authors should have added a reference for the therapeutic applications of Losartan.
5. Provide the full forms for VCID, PAMP and STZ.
6. In vitro terms should be italicized.
7. The authors should have presented results obtained from the glucose tolerance test in supplementary information.
Author Response
First, we would like to thank Reviewer Two for his valuable insights into our study. We thank the reviewer for describing our study: “Overall, the manuscript is well-written, and the results support the proposed hypothesis.” We believe the revised manuscript has improved substantially after addressing all the reviewer’s concerns. Below is our point-to-point comment on their concern.
- The authors' claim of restoring RAAS balance using Losartan, an angiotensin receptor blocker, is not novel (line 81). Angiotensin II AT1 receptor blockers as treatments for inflammatory brain disorders have already been explored.
(Saavedra JM. Angiotensin II AT(1) receptor blockers as treatments for inflammatory brain disorders. Clin Sci (Lond). 2012 Nov;123(10):567-90. doi: 10.1042/CS20120078. PMID: 22827472; PMCID: PMC3501743.)
We agree with the reviewers that our group is not the first to have neurovascular protective effects for ARBs, including Losartan. We apologize for this misunderstanding. What we meant was that our study provided new evidence that Losartan could alleviate COVID-19-induced cerebrovascular dysfunctions in diabetes.
We have reworded our manuscript and reduced our tone in line 81: “We provide evidence of restoration of RAAS balance using Losartan, an angiotensin receptor blocker, could reduce the detrimental effects of the SARS-CoV-2 spike protein in diabetes.”
- The authors should have justified their claim regarding how their mice model is a novel animal model.
As we mentioned in the manuscript, the humanized ACE2 knock-in (hACE2) mice model is successful for respiratory and viral replication studies [1-4]. To the best of our knowledge, we are the first group to use the hACE2 model for cerebrovascular complications of SARS-CoV-2 in diabetes. The manuscript has been updated as directed. Line 71:” Here we use hACE-2 to study COVID-19 cerebrovascular complications.”
- The results section is unnecessarily lengthy. Figure captions can be shortened by removing repeated statements that are already discussed in the main text. Combine the statements discussing results obtained for groups of genes expressed in response to specific molecular signaling.
We apologize for that. The manuscript is updated as directed. We removed any redundancy in the result section and the figure legends.
- The authors should have added a reference for the therapeutic applications of Losartan.
The manuscript is updated as directed.
- Provide the full forms for VCID, PAMP and STZ.
The manuscript is updated as directed.
- In vitro terms should be italicized.
The manuscript is updated as directed.
- The authors should have presented results obtained from the glucose tolerance test in supplementary information.
The manuscript is updated as directed. A new figure has been added to the supplementary material.
- Zhou B, Thao TTN, Hoffmann D, Taddeo A, Ebert N, Labroussaa F, et al. SARS-CoV-2 spike D614G change enhances replication and transmission. Nature 2021; 592(7852):122-127.
- Sun SH, Chen Q, Gu HJ, Yang G, Wang YX, Huang XY, et al. A Mouse Model of SARS-CoV-2 Infection and Pathogenesis. Cell Host Microbe 2020; 28(1):124-133 e124.
- Zheng J, Roy Wong LY, Li K, Verma AK, Ortiz M, Wohlford-Lenane C, et al. K18-hACE2 Mice for Studies of COVID-19 Treatments and Pathogenesis Including Anosmia. bioRxiv 2020.
- Zheng J, Wong LR, Li K, Verma AK, Ortiz ME, Wohlford-Lenane C, et al. COVID-19 treatments and pathogenesis including anosmia in K18-hACE2 mice. Nature 2021; 589(7843):603-607.
Round 2
Reviewer 1 Report
Comments and Suggestions for Authors
Thanks for your explanations!